# HIV and Schistosoma Co-Exposure Leads to Exacerbated Pulmonary Endothelial Remodeling and Dysfunction Associated with Altered Cytokine Landscape

**DOI:** 10.3390/cells11152414

**Published:** 2022-08-04

**Authors:** Sandra Medrano-Garcia, Daniel Morales-Cano, Bianca Barreira, Alba Vera-Zambrano, Rahul Kumar, Djuro Kosanovic, Ralph Theo Schermuly, Brian B. Graham, Francisco Perez-Vizcaino, Alistair Mathie, Rajkumar Savai, Soni Pullamseti, Ghazwan Butrous, Edgar Fernández-Malavé, Angel Cogolludo

**Affiliations:** 1Max Planck Institute for Heart and Lung Research, Member of the German Center for Lung Research (DZL), Member of the Cardio-Pulmonary Institute (CPI), 61231 Bad Nauheim, Germany; 2Institute for Lung Health (ILH), Justus Liebig University, 35305 Giessen, Germany; 3Department of Immunology, Ophthalmology and ENT, Complutense University School of Medicine and Instituto de Investigación Sanitaria Hospital 12 de Octubre (imas12), 28040 Madrid, Spain; 4Department of Pharmacology and Toxicology, School of Medicine, Universidad Complutense de Madrid and Instituto de Investigación Sanitaria Gregorio Marañón, Centro de Investigación Biomédica en Red Enfermedades Respiratorias, 28040 Madrid, Spain; 5Centro Nacional de Investigaciones Cardiovasculares (CNIC), 28040 Madrid, Spain; 6Department of Medicine, University of California, San Francisco, CA 94143, USA; 7Department of Pulmonology, I.M. Sechenov First Moscow State Medical University (Sechenov University), 119991 Moscow, Russia; 8Department of internal Medicine, Justus-Liebig University, Member of the German Center for Lung Research (DZL), 35305 Giessen, Germany; 9Medway School of Pharmacy, University of Kent and University of Greenwich, Chatham ME4 4BF, UK

**Keywords:** HIV, schistosomiasis, pulmonary arterial hypertension, inflammation, pulmonary endothelium, pulmonary vascular remodeling

## Abstract

HIV and Schistosoma infections have been individually associated with pulmonary vascular disease. Co-infection with these pathogens is very common in tropical areas, with an estimate of six million people co-infected worldwide. However, the effects of HIV and Schistosoma co-exposure on the pulmonary vasculature and its impact on the development of pulmonary vascular disease are largely unknown. Here, we have approached these questions by using a non-infectious animal model based on lung embolization of *Schistosoma mansoni* eggs in HIV-1 transgenic (HIV) mice. Schistosome-exposed HIV mice but not wild-type (Wt) counterparts showed augmented pulmonary arterial pressure associated with markedly suppressed endothelial-dependent vasodilation, increased endothelial remodeling and vessel obliterations, formation of plexiform-like lesions and a higher degree of perivascular fibrosis. In contrast, medial wall muscularization was similarly increased in both types of mice. Moreover, HIV mice displayed an impaired immune response to parasite eggs in the lung, as suggested by decreased pulmonary leukocyte infiltration, small-sized granulomas, and augmented residual egg burden. Notably, vascular changes in co-exposed mice were associated with increased expression of proinflammatory and profibrotic cytokines, including IFN-γ and IL-17A in CD4^+^ and γδ T cells and IL-13 in myeloid cells. Collectively, our study shows for the first time that combined pulmonary persistence of HIV proteins and Schistosoma eggs, as it may occur in co-infected people, alters the cytokine landscape and targets the vascular endothelium for aggravated pulmonary vascular pathology. Furthermore, it provides an experimental model for the understanding of pulmonary vascular disease associated with HIV and Schistosoma co-morbidity.

## 1. Introduction

Pulmonary vascular diseases (PVD) constitute a global health concern. Aside from the idiopathic or heritable forms, in many cases, PVD is secondary to other pathological processes affecting the pulmonary vasculature. In particular, infectious diseases such as those caused by human immunodeficiency virus (HIV) and Schistosoma are leading causes of PVD [1,2,3,4], especially pulmonary arterial hypertension (PAH) [2,5]. PAH is a progressive disease characterized by an elevation of pulmonary artery pressure (PAP) and pulmonary vascular resistance, leading to right ventricular failure and death. The elevated PAP is attributed to persistent vasoconstriction and pulmonary vascular remodeling characterized by smooth muscle cell hypertrophy and progressive neointimal proliferation of endothelial cells, leading to occlusive vascular lesions of the smallest pulmonary arteries (PA) [6,7,8]. HIV-associated PAH is characterized by proliferative vasculopathy with intimal fibrosis and the development of plexiform lesions [3]. The mechanisms involved in HIV-associated PAH are not yet well understood. Still, viral proteins have been postulated to play a role in the endothelial dysfunction observed [3,9,10,11]. On the other hand, schistosomiasis-associated lung pathology is thought to be triggered by the embolization of schistosome eggs into the lungs, leading to inflammation and pulmonary vascular remodeling. Thus, peri-egg granulomas formation and vascular remodeling with perivascular infiltrates and vessel wall thickening are considered critical events in schistosomiasis-associated pulmonary pathology [4,12,13,14]. HIV and Schistosoma co-infection is estimated in 6 million individuals worldwide [15,16]. However, to date, no clinical or experimental studies have assessed the effect of co-exposure to HIV and Schistosoma on the pulmonary vasculature. Since both Schistosoma and HIV are the main triggers of vascular pathology, it could be hypothesized that co-exposure to HIV and Schistosoma would allow for severe and rapidly pulmonary vascular disease progression, such as PAH [10]. In line with this, the requirement for a “second hit” (hypoxia, cocaine, or morphine) for HIV-associated PAH has been demonstrated in previous experimental studies [17,18,19]. Therefore, the present study was designed to define the effects of HIV and Schistosoma co-exposure on the development of vascular pathology. To this end, we made use of HIV transgenic mice harboring a replication-deficient non-infectious HIV-1 pro-viral genome [20]. This HIV transgenic line expresses seven of the nine HIV-1 proteins in different tissues, including the lung [21,22,23]. HIV transgenic mice were exposed to Schistosoma through embolization of *S. mansoni* eggs into lungs of egg-sensitized mice to model the local HIV-Schistosoma interaction and its pathological impact on the pulmonary vasculature. The results show that persistent co-exposure to HIV and Schistosoma targets the lung vascular endothelium, promoting aberrant endothelial remodeling and dysfunction associated with aggravated pulmonary vascular pathology.

## 2. Materials and Methods

### 2.1. Animals

All experimental procedures utilizing animals were carried out according to the Spanish Royal Decree 1201/2005 and 53/2013 on the Care and Use of Laboratory Animals and approved by the institutional Ethical Committees of the Universidad Complutense de Madrid (Madrid, Spain) and the Regional Committee for Laboratory Animals Welfare (Comunidad de Madrid, Ref. number PROEX-003/18). Animal studies are reported in compliance with the ARRIVE guidelines [24]. Age-matched (9–10 weeks) male FVB/NJ (Wt) and HIV-1 (HIV) transgenic mice on the FVB/NJ background (FVB/N-Tg(HIV)26 Aln/PkltJ; Tg26) from the Jackson Laboratory (USA) were provided by Charles River (France). This HIV-1 Tg26 mice model expresses a transgene containing a portion of the HIV genome, including Env and Tat, Nef, Rev, Vif, Vpr, and Vpu accessory genes but lacking part of the gag-pol region, rendering the virus non-infectious [20]. Animals were kept under standard conditions of temperature 22 ± 1 °C and 12:12 h dark/light cycle with free access to food and water.

### 2.2. Treatment with Schistosoma Eggs

Mice were randomly assigned to four groups: Wt, parasite egg-treated Wt (Wt+Schisto), HIV, and egg-treated HIV (HIV+Schisto). *Schistosoma mansoni* eggs were isolated from homogenized and purified livers of Swiss-Webster mice infected with cercariae, provided by the Biomedical Research Institute (Rockville, MD, USA). Eggs were then inactivated by freezing (−60 to −80 °C) and stored at −80 °C until used. To induce pulmonary vascular disease, we used an experimental model previously reported [25,26,27,28]. In brief, mice were intraperitoneally sensitized to 240 *Schistosoma mansoni* eggs/gram body weight and then intravenously challenged two weeks later with 175 *Schistosoma mansoni* eggs/gram body weight. Animals were analyzed seven days after intravenous egg administration (Appendix A). Control mice received the same volume of 1.2% sodium chloride used as a vehicle. Experiments were performed in a coded format, with the investigators lacking knowledge of the specific experimental group.

### 2.3. Hemodynamic Measurements

One week after intravenous administration of *S. mansoni* eggs, mice were anesthetized i.p. with a mixture of 80 mg/kg ketamine (Mearial, Lyon, France) plus 8 mg/kg xylazine (KVP Pharma und Veteriär-Produkte GmbH, Kiel, Germany). Before initiation of the surgical procedure, general anesthesia was confirmed by assessing the absence of response to any stimulus. Then, animals were placed in a supine position on a thermostatically controlled electric heating blanket (Homeothemic Blanket Control Unit, Harvard Apparatus, March-Hugstetten, Germany) to maintain body temperature at 38 °C. The tracheostomy was performed by a ventral neck incision followed by the insertion of a 1.3 mm outer diameter tracheotomy cannula in the trachea. Animals were ventilated with room air (tidal volume 9 mL/kg, 100 breaths/min, and a positive end-expiratory pressure of 2 cm H_2_O) with a rodent ventilator (MiniVent Type 845, Harvard Apparatus, MA, USA). After sternotomy, right ventricular systolic pressure (RVSP) and systolic, diastolic, and mean PAP (sPAP, dPAP, and mPAP) were measured in open-chest mice as previously reported [11]. Measurements were recorded with a pressure transducer via a 0.7 mm internal diameter catheter (24 GA, BD Insyte, CA, USA) introduced in the right ventricle and then advanced to the main PA. Thereafter, animals were sacrificed by exsanguination in the continuous presence of anesthesia, and organs were harvested for analysis.

### 2.4. Cardiac Remodeling

At the end of the recordings, hearts were excised, and the right ventricle (RV) and the left ventricle plus septum (LV+S) were carefully dissected and weighed. The Fulton index [RV/(LV+S)] and the ratio RV/body weight (BW) were calculated to assess right ventricular hypertrophy [11].

### 2.5. Recording of Pulmonary Arterial Vasodilation

Resistance PAs were carefully dissected free of surrounding tissue, cut into rings (1.8–2 mm length), and placed in a sterile plate containing serum-free DMEM for 20 h. After that, PA rings were mounted in a wire myograph with Krebs buffer solution maintained at 37 °C and bubbled with 21% O_2_, 74% N_2,_ and 5% CO_2_ [11]. Vessels were stretched to give an equivalent transmural pressure of 20 mmHg. Preparations were firstly stimulated by raising the K^+^ concentration of the buffer (to 80 × 10^−3^ M) in exchange for Na^+^. Vessels were washed three times and allowed to recover before a new stimulation. The relaxant effects induced by acetylcholine (ACh, 10^−9^–10^−5^ M, and endothelial-dependent vasodilator) or sodium nitroprusside (SNP, 10^−11^–10^−5^ M, an endothelial-independent vasodilator) were examined in arteries stimulated with serotonin (5-HT, 10^−5^ M).

### 2.6. Isolation of Lung Leukocytes and Flow Cytometry

After measuring PAP, mouse lungs were perfused by right ventricle administration of PBS, dissected, and digested as previously described [28]. Briefly, left lobes were digested with Liberase TM (Roche Diagnostics GmbH, Roche Applied Science, Mannheim, Germany), disrupted, and filtered using a 40 µm cell strainer (Corning^®^, Somerville, NY, USA), followed by centrifugation for 10 min at 300× *g*. Red blood cells were lysed with ACK lysis buffer, and cells were resuspended in RPMI medium (Gibco, Thermo Fisher Scientific, Inc., Waltham, MA, USA), filtered again, centrifuged, and resuspended in RPMI. Cell counting was performed using a Neubauer chamber after staining with trypan blue to exclude dead cells. For flow cytometry analysis, cells were resuspended in FACS buffer (PBS containing 5% bovine serum albumin and 0.1% sodium azide). Following the blocking of Fc receptors with anti-CD16/CD32 antibodies (BD Biosciences, San Jose, CA, USA), cells were stained with fluorochrome-conjugated antibodies to extracellular CD45, CD4, CD3, TCRβ, TCRδ, and intracellular staining for IL-6, IL-13, IL-17, and IFN-γ, all from BD Biosciences. Data were acquired with a BD FACSCalibur flow cytometer and analyzed using FlowJo™ v10 Software (BD Biosciences, FlowJo, LLC, Ashland, OR, USA) following the gating strategy as indicated in Appendix A.

### 2.7. Schistosoma Egg Counting

The number of *S. mansoni* eggs present in the mouse lung tissue was determined after digestion of a piece of right lung lobe with shaking in 4% potassium hydroxide for 18 h at 33 °C. The number of eggs present in aliquots of the digest was counted three times, as previously described [13].

### 2.8. Lung Histology

The right lung was washed with a saline solution followed by a 4% paraformaldehyde infusion through the right bronchus and embedded in paraffin. All sections were cut at 5 µm, stained with hematoxylin and eosin, and examined by light microscopy. For quantification of pulmonary vascular remodeling, PA (25–250 µm outer diameter; OD) was analyzed in a blinded fashion and categorized as muscular, partially muscular, or non-muscular. The medial wall thickness was examined by light microscopy, and elastin was visualized by its green autofluorescence. For vessel occlusion analysis, all small pulmonary vessels per cross-section of the right lobe were evaluated and classified according to the evidence of luminal occlusion. Around 500 representative vessels within a range of diameters from 20 to 70 μm were measured per sample. Peri-egg granulomas size was determined in histological sections stained with hematoxylin and eosin containing a single visible egg using NDP view2 software (Hammamatsu, Japan).

### 2.9. Immunohistochemical Analysis

Paraffin sections of 3-μm thickness were stained with antibodies to α smooth muscle actin (α-SMA) (dilution 1:900, clone 1A4, Sigma, St. Louis, MO, USA), von Willebrand factor (vWF) (dilution 1:900, Dako, Hamburg, Germany) and proliferating cell nuclear antigen (PCNA) (dilution 1:200, sc-56, Santa Cruz Biotechnology, Inc., Santa Cruz, CA, USA). Automated quantification of PCNA immunopositive labeling was performed using QuPath [29]. The software was trained to recognize PCNA-stained nuclei and positive-labeled cells using positive cell and subcellular detection modules. The number of cells with PCNA-positively labeling per μm^2^ and the percentage of cells detected with PCNA immunolabeling were recorded and compared between all groups.

### 2.10. Classification of Occluding Lesions

Small arteries with an outer diameter (OD) <100 μm were analyzed, and the type of vessel occlusion was determined according to the pattern of von Willebrand factor (vWF) staining. Briefly, non-plexiform-like lesions presented vWF-positive (endothelial) cells as a concentric rim of cells in the inner layer, while plexiform-like lesions showed a complex/disorganized luminal occlusion [30].

### 2.11. Immunofluorescence Staining and Analysis

Paraffin-embedded lung tissue sections (3-μm thick) were dewaxed and rehydrated. Antigen retrieval was performed by pressure cooking of lung sections in citrate buffer (pH 6.0) (Ref 005000, Thermo Scientific, Thermo Fisher Scientific, Inc., Waltham, MA, USA) and blocked with 5% BSA solution. Slides were incubated with different primary antibodies overnight at 4 °C. After overnight incubation, slides were washed and incubated with the respective secondary antibodies for 1 h. For triple and quadruple staining, the same protocol was repeated for primary and secondary antibodies on consecutive days. All sections were counterstained with DAPI (Ref 62248, Thermo Scientific, Thermo Fisher Scientific, Inc., Waltham, MA, USA) and mounted with a water-based mounting medium (Fisher Scientific). For TUNEL, the In Situ Cell Death Detection Kit (Roche Diagnostics GmbH, Roche Applied Science, Mannheim, Germany) was used following the manufacturer’s recommendations. Antibodies to PCNA and ERG were from Sigma-Aldrich and Abcam, respectively.

### 2.12. Assessment of Collagen Deposition

Collagen deposition in the lung was measured by Sirius red staining. Paraffin-embedded lung sections were preheated at 63 °C for 30 min. Paraffin was removed with xylene followed by serial rehydration with decreasing percentages of ethanol and washed for 2 min in distilled water. The sections were placed in picrosirius red (Sigma) solution for 60 min and in acidified water (acetic acid in distilled water) for 4 min, followed by two washes, first in ethanol and then in xylene. Finally, the slides were mounted with ProLong™ Gold Antifade Mountant (Life Technologies, Thermo Fisher Scientific, Inc., Massachusetts, USA). Quantification of the percent vascular area fraction positive for Sirius red staining was performed with ImageJ.

### 2.13. Protein Expression

Mouse lung samples were homogenized in a buffer containing 0.2 M Tris pH = 7.5, 1 mM DTT (Sigma-Aldrich), 1% Nonidet P40, 1% phosphatase, and protease inhibitors, all from Roche, and a TissueLyser II (QIAGEN). For Western blot, 30 μg of protein was used for each condition. Protein samples were loaded in 7.5% acrylamide/bis-acrylamide SDS-PAGE gels and subsequently subjected to immunoblotting. eNOS expression was detected using an anti-eNOS antibody (1:1000, BD Bioscience) and a peroxidase-coupled anti-mouse secondary antibody (1:10,000, Sigma-Aldrich). Β-actin was used as a loading control (1:10,000, Sigma-Aldrich). Blots were analyzed using an Odyssey Fc System (LiCOR Bioscience, Lincoln, NE 68504, USA) and quantified using ImageJ software (National Institutes of Health, Rockville, MD, USA; http://imagej.net/ImageJ, 19 May 2022).

### 2.14. Reagents

Drugs and reagents were obtained from Sigma-Aldrich Quimica (Madrid, Spain). Drugs were dissolved in distilled water.

### 2.15. Statistical Analysis

Data are expressed as mean ± SEM.; n indicates the number of experiments from different animals unless otherwise stated. Statistical analysis was performed using Student’s *t*-test and one-way ANOVA (for normally distributed data) followed by Tukey’s post-hoc test or non-parametric Kruskal–Wallis test. Two-way ANOVA and the Bonferroni multiple comparison test were used to compare dose–response curves. When more than one sample came from the same animal, the nested ANOVA was applied. Differences were considered statistically significant when the *p*-value was less than 0.05.

## 3. Results

### 3.1. HIV and Schistosoma Co-Exposure Associates with Augmented PAP and RV Mass in the Absence of Overt PAH and Cardiac Hypertrophy

We analyzed the impact of individual and combined HIV and Schistosoma exposure on pulmonary hemodynamics via right heart catheterization (Figure 1A–D). Untreated Wt and HIV mice had similar RVSP, systolic, diastolic, and mean PAP. Egg-treated mice showed a general tendency to higher values of these hemodynamic parameters, with the differences being significant for diastolic and mean PAP in egg-treated HIV mice compared to untreated ones. Noticeably, egg-treated HIV mice demonstrated the highest values for all the hemodynamic parameters analyzed.

We also examined the effects of HIV and Schistosoma co-exposure on cardiac remodeling. Surprisingly, we found a reduced Fulton index in HIV animals (Appendix A), which could be associated with a modest LV hypertrophy displayed by untreated and egg-treated HIV animals compared to Wt counterparts (Appendix A). However, when expressed in terms of body weight (Appendix A), HIV animals had similar RV values as untreated or egg-exposed Wt mice, while a slight but significant increase was observed in egg-treated HIV compared to Wt mice (Appendix A). Thus, in HIV and Schistosoma co-exposed mice, pulmonary vascular remodeling was associated with a consistent increase in PAP but without overt PAH or RV hypertrophy.

### 3.2. Pulmonary Endothelial Function Is Suppressed in HIV Mice Exposed to Schistosoma Eggs

Next, we analyzed endothelial-dependent relaxation assessed following the addition of ACh to isolated PA previously stimulated with serotonin (Figure 2A–D). The relaxation induced by ACh (10^−6^ M) was moderately attenuated in egg-treated Wt (16 ± 7%; *p* < 0.01) and untreated HIV (20 ± 1%; *p* < 0.05) mice as compared to Wt mice (37 ± 2%, Figure 2A–C,E). Remarkably, PA from egg-exposed HIV mice showed negligible endothelial-dependent vasodilation, even at the highest concentrations of ACh tested (Figure 2D,E). In fact, the area under the curve of Ach-induced relaxation confirmed the impairment of endothelial-dependent vasodilation in egg-exposed HIV mice (Appendix A). Consistent with our functional data, we found that the expression of eNOS, the main source of endothelial NO, was markedly reduced in the lungs of co-exposed mice compared to Wt (Figure 2F). In contrast, the endothelial-independent relaxation induced by SNP (Appendix A) was comparable in untreated and egg-treated Wt and HIV mice. We also assessed if pulmonary vascular dysfunction in co-exposed mice was associated with increased apoptosis by examining TUNEL staining in lung sections. A clear TUNEL-positive staining was observed in the inner, but not in the media, layer of PA from egg-treated HIV mice, while this was not found in Wt mice (Figure 2G). Taken together, these findings suggest that combined exposure to HIV and Schistosoma impact the functional properties of the pulmonary vasculature mainly by targeting the vascular endothelium.

### 3.3. Schistosoma Eggs Exposure Increases Medial Wall Thickness of Small Pulmonary Arteries in Both Wt and HIV Mice

To determine the impact of individual or combined HIV and Schistosoma exposure on pulmonary vasculature morphology and remodeling, we analyzed hematoxylin/eosin-stained lung sections from untreated and egg-treated Wt and HIV mice (Appendix A). Small PAs were classified in a blinded fashion as muscular, partially muscular, and non-muscular arteries. We found a modest vascular remodeling in HIV compared to Wt mice, suggested by an increased percentage of partially muscular PA and a decreased percentage of non-muscular PA (Figure 3A). These observations are in line with our previous study [11]. Administration of Schistosoma eggs in Wt mice led to a clear vascular remodeling indicated by a much higher percentage of muscular PA and a lower percentage of non-muscular PA. A similar pattern of PA muscularization was observed in egg-exposed HIV mice. Quantification of the vascular remodeling in PA (250–50 μm) showed a comparable increase in the medial wall thickness in egg-exposed Wt and HIV mice, compared respectively to untreated counterparts (Figure 3B), with this enhancement being not significantly different between egg-treated mice.

### 3.4. Co-Exposure to HIV and Schistosoma Eggs Increases Pulmonary Endothelial Proliferation, Vessel Obliteration, and Formation of Plexiform-like Lesions

Histological examination of lung sections demonstrated enhanced PCNA staining in egg-treated compared to untreated mice (Appendix A). Specific quantification of PCNA-positive cells within the vessel’s area showed a significant increase in animals exposed to Schistosoma eggs (Appendix A). Of note, egg-exposed HIV mice exhibited significantly higher frequencies of proliferating cells compared to all other groups. Moreover, in stained lung sections from egg-treated animals, occluded vessels were strongly positive not only for PCNA expression (Figure 3C, left) but also for the endothelial cell marker von Willebrand factor (vWF) (Figure 3C, right). Noticeably, these endothelial cells did not form a monolayer, as expected in healthy vessels, but piled up to fill the vascular lumen, suggesting the involvement of endothelial cell proliferation in vessel obliteration. To gain more information on this, we performed a PCNA, α-SMA (as a marker of smooth muscle cells), and ERG (as a marker of endothelial cells) immunofluorescence co-staining in lung samples from co-exposed animals. Representative images in Appendix A show that while non-occluded vessels from Wt or co-exposed animals are essentially PCNA-negative, partially occluded vessels in HIV+Schisto mice showed high PCNA-positive staining, which corresponds to endothelial and other unidentified cells (probably immune cells) but not to smooth muscle cells.

Analysis of fully closed vessels revealed that none of the untreated Wt or HIV mice showed lumen occlusion of pulmonary vessels. In contrast, occluded vessels were observed in mice exposed to Schistosoma eggs (Appendix A). Notably, 9 out of the 19 egg-exposed Wt animals (47%) displayed at least one occluded vessel, while this was observed in 14 out of 20 (70%) in the egg-exposed HIV group (Appendix A). Indeed, the percentage of occluded vessels was double in egg-treated HIV mice compared to Wt counterparts (Figure 3D).

Additionally, we studied the type of intimal remodeling in these occluding lesions as described in Material and Methods. Non-plexiform and plexiform-like lesions were observed in small pulmonary vessels from both egg-exposed Wt and HIV mice (representative images in Figure 3E). However, egg-treated HIV mice developed severe occlusive lesions with a plexiform-like phenotype more frequently compared to their Wt counterparts (Figure 3F). Taken together, these results suggest that co-exposure to HIV and Schistosoma exacerbates endothelial cell proliferation and the formation of plexiform-like lesions associated with faster and enhanced occlusion of small pulmonary vessels.

### 3.5. HIV Mice Display Pulmonary Perivascular Fibrosis That Is Exacerbated by Exposure to Schistosome Eggs

We further explored the impact of individual or combined HIV and Schistosoma exposure on the pulmonary vasculature and assessed the perivascular collagen deposition by Sirius red staining, as shown by representative images in Figure 4A. Quantification of Sirius red staining (Figure 4B) revealed that pulmonary vessels from HIV mice had a significantly higher degree of fibrosis compared to Wt animals. Administration of parasite eggs resulted in augmented collagen deposition in vessels from both mice, although of a greater magnitude in HIV mice. Noticeably, pulmonary vessels from untreated HIV mice demonstrated levels of fibrosis similar to those in egg-treated Wt counterparts (Figure 4B). These findings suggest that HIV mice have an intrinsic tendency to develop pulmonary vascular fibrosis, which could be further enhanced by schistosome eggs trapped in the lung vasculature.

### 3.6. Altered Pulmonary Immune Response to Schistosoma Eggs in HIV Mice

Leukocyte infiltration into the lung and formation of peri-egg granulomas are typical immune responses induced by Schistosoma eggs trapped in the pulmonary vasculature [13,26]. Flow cytometry analyses in cells isolated from the lungs of Wt and HIV mice (Figure 5A) showed comparable frequencies of leukocytes, which remained unaltered after administration of *S. mansoni* eggs. Leukocyte numbers (Figure 5B) were also comparable in untreated Wt and HIV mice, but with a tendency for higher leukocyte counts in the latter animals (Wt 1.54 × 10^6^ ± 0.28 × 10^6^; HIV 2.97 × 10^6^ ± 0.90 × 10^6^). Upon Schistosoma egg administration, leukocyte numbers were significantly increased in Wt but not in HIV mice (Figure 5B).

Embolization of Schistosoma eggs also led to the formation of peri-egg granulomas in the lung of Wt and HIV mice, which displayed a similar gross appearance in both mice (Figure 5C). The number of granulomas was also comparable between Wt and HIV mice (Figure 5D), but, interestingly, granuloma size was significantly smaller in the latter (Figure 5E). Moreover, granulomas in HIV mice showed an impaired capability for egg clearance, as indicated by the significantly higher residual egg burden observed in these mice compared to Wt counterparts (Figure 5F). Taken together, these results suggest that the granulomatous response induced by Schistosoma eggs in the lung is impaired in HIV mice.

### 3.7. HIV and Schistosoma Co-Exposure Alters the Cytokine Landscape in the Lung

For characterization of the pulmonary cytokine landscape of Wt and HIV mice, untreated or treated with schistosome eggs, a combination of immunofluorescence microscopy and intracellular flow cytometry was used for identification of type 1 (IFN-γ), type 2 (IL-4, IL-13) and type 17 (IL-17A) cytokines. Intracellular cytokine expression in pulmonary T cells and myeloid cells by flow cytometry was determined following the strategies depicted in Appendix A.

Type 2 cytokines (i.e., IL-4 and IL-13) are typically associated with the granulomatous response to parasite eggs in schistosomiasis. Pulmonary IL-4 expression was hardly detected in untreated Wt and HIV mice but strongly induced in granulomas of egg-treated mice (Figure 6A), and mostly in perivascular areas and to a lesser degree in intravascular locations. Quantification of the fluorescent cytokine signal confirmed these results (Figure 6B). Intracellular flow cytometry analysis (Figure 6D) showed that in untreated mice, either Wt or HIV, IL-4 is mainly expressed by CD4^+^ T cells. After egg administration, the relative abundance of IL-4^+^ CD4 T cells was reduced in both mice, with a concomitant increase in that of IL-4^+^ myeloid cells.

In the case of IL-13, this cytokine was readily detected in the lung of untreated HIV but not in Wt mice (Figure 6A, lower panels). Egg treatment resulted in a markedly increase in IL-13 in HIV mice and less prominently in Wt counterparts. Quantification of the IL-13 signal confirmed these observations (Figure 6C). Of note, in HIV but not Wt mice, IL-13^+^ cells were detected outside but in close proximity to the vessel wall. Cells in this location were increased in egg-treated HIV mice, along with IL-13^+^ cells in the periphery of the granuloma. In contrast, in egg-treated Wt animals, IL-13^+^ cells were present only in the outer areas of the granuloma (Figure 6A, lower panels). Analysis of pulmonary IL-13 expression by intracellular flow cytometry (Figure 6E) showed an increased relative abundance of IL-13-expressing CD4 T cells in untreated HIV mice compared to Wt counterparts. Upon parasite egg administration, these cells were increased (but not significantly) in Wt mice and decreased significantly in HIV mice. Analysis of the myeloid compartment revealed an increased frequency of IL-13^+^ cells in egg-treated HIV mice compared to the other groups (Figure 6E). Collectively, these results reveal that IL-4 and IL-13 expression are differentially affected in pulmonary T and myeloid cells of HIV after parasite egg administration.

Regarding type I cytokines, immunofluorescence staining of lung sections revealed a clear IFN-γ expression in HIV but not in Wt mice. Egg administration increased IFN-γ expression in perivascular granulomas in both mice but seemingly more in HIV mice (Figure 7A, upper panels), which, in contrast to Wt counterparts, showed not only perivascular but also markedly intravascular IFN-γ expression. Quantification of the cytokine signal confirmed this augmented IFN-γ expression in HIV compared to Wt mice, either untreated or treated with parasite eggs, but particularly in the latter (Figure 7B). Flow cytometric analysis revealed that CD4 T cells and γδT cells in HIV mice were the main cell types involved in the augmented IFN-γ response to parasite eggs, with a lesser contribution of myeloid cells (Figure 7C).

Type 17 responses were assessed by measuring pulmonary IL-17A expression. Immunofluorescence analysis showed a very low abundance of proinflammatory IL-17A-expressing cells in the lung of Wt and HIV mice (Figure 7A, lower panel). Notably, these cells were strongly augmented in perivascular granulomas of both mice but more markedly in HIV mice, with IL-17A^+^ cells being detected both around and within the vessel. The increased expression of IL-17A by egg exposure, especially in the lungs of HIV mice, was confirmed by flow cytometry (Figure 7D). Egg-induced augmentation of pulmonary IL-17A expression was associated mainly with γδ T cells in Wt mice, while CD4^+^ and γδ T cells contributed both to the increased expression of IL-17A in HIV mice (Figure 7E); with just a minor contribution from the myeloid compartment in both mice (Figure 7E).

## 4. Discussion

HIV and Schistosoma infections represent two major causes of morbidity and mortality globally. Among their different detrimental effects, both diseases may target the lung and contribute to the development of pulmonary vascular diseases [10,31]. Herein we report for the first time the impact of HIV and Schistosoma co-exposure on the pulmonary vasculature and the development of pulmonary vascular pathology by using a non-infectious mouse model based on HIV (Tg26) transgenic mice subjected to sensitization and embolization with *S. mansoni* eggs. Our data showed that HIV and Schistosoma co-exposure results in suppressed endothelial-dependent vasodilation and an increased number of occluded pulmonary vessels with plexiform-like lesions, with augmentation of PAP. These pathological changes were associated with remarkable changes in the expression of proinflammatory and profibrotic cytokines in the lungs of co-exposed mice. In contrast, medial thickening or cardiac remodeling was not aggravated by combined exposure to HIV proteins and schistosome eggs.

We found that neither exposure to parasite eggs nor HIV proteins alone affected pulmonary or RV hemodynamics and that only the exposure to both led to a clear but not overt increment in PAP and RVSP. Similarly, previous studies in other HIV models have found a requirement for a “second hit”, such as cocaine [18], opioids [17], or hypoxia [19], to elevate pulmonary pressure or exacerbate the density of pulmonary lesions [32]. In addition, only modest hemodynamic changes were observed in animals with abundant pulmonary vascular lesions due to endothelial cell apoptosis [33]. Moreover, in the hypoxia+Sugen5416 model, restoration of endothelial shear responsiveness was shown to reverse intimal occlusions but with minor consequences on medial thickening or RVSP [34].

Endothelial dysfunction represents a common hallmark in HIV-infected patients regardless of antiretroviral treatment [3,10] and is considered a precursor of the development of HIV-associated cardiovascular and lung diseases that contribute substantially to morbidity and mortality [3,9,10,35]. In agreement with our previous work [11], herein, we observed that pulmonary vessels from HIV mice displayed endothelial dysfunction, which was greatly exacerbated by exposure to schistosome eggs. Among the potential underlying mechanisms, HIV proteins, particularly Nef, Tat, and gp120, appear to be involved as they have been shown to play a key role in the development of endothelial injury [3,9,10,35,36,37,38,39]. Of note, Chelvanambi et al. [40] have recently reported that Nef protein persists in the lungs of HIV patients on antiretroviral therapy, causing endothelial apoptosis and pulmonary vascular damage.

It is also widely accepted that endothelial dysfunction is a common initiating event that contributes to several forms of chronic lung disease, such as PAH [7,41]. Indeed, endothelial dysfunction is considered an initial trigger for a number of functional and histopathological features shared in most types of PAH, such as increased vasoconstriction, muscularization, intima and media thickening, vessel obliteration, and complex plexiform lesions [6,8,30]. In the present study, we show that pulmonary vessels from HIV transgenic mice display moderate endothelial dysfunction but only mild muscularization and no evidence of medial thickening or vessel occlusion. Similar results have been reported in other HIV experimental models [17,19]. In contrast, exposure of HIV mice to schistosome eggs led to a marked suppression of endothelial-dependent vasodilation, indicating an exacerbation of the endothelial injury. This was associated with blunted eNOS expression and with increased apoptosis in the inner layer of PA from co-exposed animals. *S. mansoni* egg products have been shown to be detrimental to endothelial cells by inducing the release of IL1-β from immune cells, which in turn causes endothelial cell dysfunction and apoptosis [42]. However, how parasite eggs combine with HIV proteins to exacerbate pulmonary endothelial dysfunction remains to be fully determined.

Furthermore, endothelial damage is critically involved in the development of obliterative pulmonary vascular lesions [7,8,33]. Remarkably, we found that pulmonary vessels became occluded with proliferating endothelial cells more frequently in egg-treated HIV compared to Wt mice. The formation of plexiform lesions represents a morphological hallmark of pulmonary arteriopathy in severe PAH [6,8]. While the cellular and molecular mechanisms have not been fully elucidated, the current hypothesis postulates that the lesion is initiated by endothelial apoptosis followed by the proliferation of apoptosis-resistant endothelial cells [7,8,33,43]. In line with this notion, we observed increased apoptosis in the tunica intima of non-occluded vessels from egg-treated HIV mice, which suggests that endothelial apoptosis may be the initial trigger for vessel obliteration. Moreover, vessel occlusion in co-exposed animals was associated with enhanced proliferation of endothelial but not smooth muscle cells. Remarkably, using an approach similar to that of Toba et al. [30], we detected a much higher proportion of plexiform-like lesions in egg-treated HIV than in Wt mice. This is one novel finding very relevant to the pathophysiology of pulmonary vascular disease, as it indicates that combined exposure to HIV and Schistosoma results in the early onset of pulmonary lesions that normally occurs at advanced and severe stages of PAH [30]. In fact, plexiform-like lesions have not been consistently found in rodent models of PAH [44]. While the mechanisms triggering these lesions are still unknown, it is tempting to speculate that HIV proteins such as Nef, Tat, and gp120 may be involved [3,9,10,35]. Further supporting this notion, complex pulmonary vascular lesions were observed in macaques infected with chimeric simian/human immunodeficiency virus (SHIV) containing Nef and in patients with HIV-related pulmonary hypertension [45,46].

In HIV transgenic mice, viral transcripts are expressed at low levels in immune cells such as monocytes, macrophages, and lymphocytes [21], similarly to HIV-infected people treated with ART [3,35]. Thus, HIV-related alterations in the population of lung resident immune cells [3,47] could compromise the initiation of responses to local inflammatory stimuli [48], such as embolized parasite eggs. In addition, impaired pulmonary recruitment of circulating inflammatory cells could affect the progression of egg-induced immune responses in HIV mice, resulting in defective growth and egg-clearing activity of granulomas [28,49]. Supporting this notion, we observed that HIV mice showed impaired leukocyte infiltration in response to parasite eggs and a clear deficit in granuloma growth and function. Of note, impaired macrophage infiltration into the lung and delayed resolution of lung inflammation has been reported recently in HIV Tg26 transgenic mice in response to bacterial lipopolysaccharide, with these defects being corrected by inhibition of HIV transcription [50]; suggesting an association between HIV gene expression in leukocytes and their impaired recruitment to sites of inflammation. Thus, in egg-treated HIV mice, pulmonary persistence of parasite eggs [12,28] and HIV proteins could, on the one hand, cause direct damage to the lung vasculature [9,40,51]. On the other hand, HIV proteins and schistosome egg products could sustain a chronic inflammatory process targeting the vasculature [3,14,35,41]. In line with the latter, we found that HIV and Schistosome co-exposure caused enhanced expression of the proinflammatory cytokines IFN-γ and IL-17A in the lung tissue.

IFN-γ was increased to a large extent in perivascular and intravascular locations of perivascular granulomas in egg-treated HIV mice, with CD4^+^ and γδ T cells and, to a lesser extent, myeloid cells (likely macrophages according to their morphology), being the main cellular sources. Heightened IFN-γ expression in pulmonary γδ T cells was atypical, as the normal predominant population of γδ T cells in the lung is IL-17-producing cells [52].

Increased abundance of pulmonary IFN-γ in egg-treated HIV mice could have indirect but relevant effects on local antigen presentation and subsequent activation of CD4^+^ T cells, not only by macrophages but also by fibroblasts with heightened levels of MHC class II molecules [53]. In addition, IFN-γ, along with IL-8, has been shown to be an inflammatory marker of pulmonary vascular disease, particularly in PAH [54]. Moreover, levels of IFN-γ were elevated in sera from patients with systemic sclerosis-associated PAH [55]. In addition, IFN-γ promoted vascular remodeling in human microvascular endothelial cells by upregulating endothelin (ET)-1 and TGF-β [56].

With regard to IL-17A expression in the lung, immune cells expressing this cytokine were augmented in egg-induced perivascular granulomas of both HIV and Wt mice but more markedly in HIV mice, both around and within vessels. Of note, CD4^+^ and γδ T cells were the main producers of IL-17A in egg-treated HIV mice, while in Wt counterparts, only γδ T cells were augmented. This suggests that the combined presence of HIV proteins and schistosome eggs in the lung recruits and stimulates a larger variety of IL-17A-producing cells compared to individual exposure. In line with this, IL-17 was induced in Th, NKT, and γδ T cells following PMA/ionomycin stimulation, but γδ T cells exhibited the largest increase in expression [57].

Th17 effector cells can be induced in parallel to Th1 cells, and both types colocalize regionally and may require each other for recruitment into the region [58]. This is consistent with our data showing abundant Th1 and Th17 cells in perivascular granulomas of egg-treated HIV mice and notably inside occluded vessels.

In experimental animal models of hypoxia-induced PH, Th17 cells have been shown to play a pathogenic role [59,60]. Moreover, purified CD4^+^ T cells from PAH patients expressed a higher level of IL-17 after activation than did those from control subjects [61]. Furthermore, in a mouse model of silica-induced lung inflammation and fibrosis, it was found that IL-17A production by Th17 and γδ T cells was required for early lung neutrophilic inflammation and acute tissue injury [62]. Also of note, T cells and γδ T cells, in particular, have been identified as proinflammatory and profibrotic mediators in the initiation and progression of pulmonary fibrosis, partly by producing IL-17 [63]. Indeed, reducing IL-17 activity using anti-IL-17A antibodies decreased infiltration of inflammatory cells and collagen deposition in the liver of schistosome-infected mice [57].

In regard to type 2 cytokines, global IL-4 expression was similarly upregulated in the lung of egg-treated HIV and Wt mice, while IL-13 was more markedly increased in egg-treated HIV mice than in Wt counterparts. Notably, in our model, these cytokines were derived mostly from myeloid cells, where IL-13 but not IL-4 was strongly enhanced in co-exposed mice. In contrast, IL-13 expression was significantly downregulated in CD4^+^ T cells from egg-treated HIV mice. Although the precise identity of IL-13-expressing myeloid cells is still to be clearly established, their size, morphology, and location within the granuloma are consistent with their being macrophages. Taken together, these results indicate that the expression of HIV proteins in distinct subsets of pulmonary immune cells, i.e., myeloid versus T cells, differentially alter the induction of IL-13 responses to parasite eggs. Further, they suggest that HIV protein expression in CD4^+^ T cells could affect their differentiation in Th2 effectors, which are critical at the beginning of the parasite egg-induced granulomatous response [28,35], leading ultimately to pulmonary vascular pathology and PH [27,64,65].

IL-13 and IL-17 are endowed with potent profibrotic capacity with relevance to lung fibrosis [63,66,67]. We found that pulmonary vessels from HIV mice exhibited increased perivascular fibrosis, which was further enhanced by lung embolization of schistosome eggs. In line with this, a recent study reported a profibrotic phenotype of primary lung fibroblasts isolated from HIV (Tg26) transgenic mice and in wild-type primary lung fibroblasts cultured with HIV-1 protein gp120 [68]. Because schistosomiasis and HIV infection have both been associated with lung fibrosis [4,69], our findings suggest that combined exposure to HIV and Schistosoma could increase the risk of developing pathological fibrotic changes in the pulmonary vasculature by mechanisms partly involving dysregulated expression of IL-13 and IL-17.

In our model, unlike some previous studies [25,27,28], lung embolization of schistosome eggs was not associated with overt pulmonary hypertension or cardiac remodeling. Other studies have also found no significant elevation of RVSP or RV hypertrophy in mouse models of experimental schistosomiasis [12,13] even when, as in our study, an increased muscularization of small pulmonary vessels, thickening of the medial layer, and even occasional plexiform-like lesions could be observed [12]. Differences in experimental conditions such as altitude (Denver vs. Madrid) or mouse genetic background (C57BL/6 vs. FVB) [70] could explain discrepancies among studies.

Our study has the limitation of examining the co-exposure to HIV and Schistosoma only in males. Although the administration of *Schistosoma mansoni* eggs seems to induce a comparable immune response with an associated granuloma formation and pulmonary vascular remodeling in male [71] and female [14,28] mice, it would be interesting to evaluate if combined exposure to HIV and Schistosoma exerts a differential impact in females as compared to the present study in males.

Our non-infectious mouse model of HIV and Schistosome co-exposure, in contrast with other models, avoids the systemic effects associated with infection that could confound the interpretation of results [72]. In addition, it precludes potential misleading effects due to the interaction of live HIV with a live parasite, for instance, the integration of HIV-1 in the genome of *S. mansoni* [73]. From the technical point of view, our model circumvents the laborious protocols and the variability of the outcome typically associated with reconstitution approaches, such as humanized mice [3].

Schistosomiasis and HIV infection represent two of the most common causes of pulmonary vascular pathology worldwide. Due to their high incidence globally and in endemic areas, it is conceivable that many individuals are co-infected. In the present study, we describe for the first time the impact of combined HIV and Schistosoma exposure on the pulmonary vasculature, and provide a novel insight into the structural and functional alterations occurring as a result of such HIV-Schistosoma interaction. Our work suggests that persistent expression of HIV proteins in the lungs, as occurs in ART-treated HIV-infected people, may cause an initial endothelial insult that may be aggravated in a subsequent Schistosoma infection, partly by induction of a proinflammatory and pro-fibrotic cytokine milieu. Together, this may ultimately foster the development of pulmonary vascular pathology.

## Figures and Tables

**Figure 1 cells-11-02414-f001:**
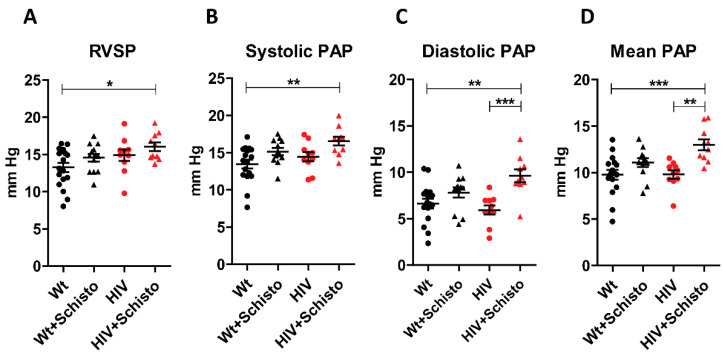
**Schistosoma egg exposure increases pulmonary arterial pressure in HIV mice**. Mean values of (**A**) right ventricular systolic pressure (RVSP), (**B**) systolic, (**C**) diastolic, and (**D**) mean pulmonary arterial pressure (PAP) in Wt and HIV mice exposed or unexposed to Schistosoma. Results are expressed as means ± SEM (n = 10–17). *, ** and *** indicate *p* < 0.05, *p* < 0.01, and *p* < 0.001, respectively (one-way ANOVA analysis followed by the Tukey post-hoc test).

**Figure 2 cells-11-02414-f002:**
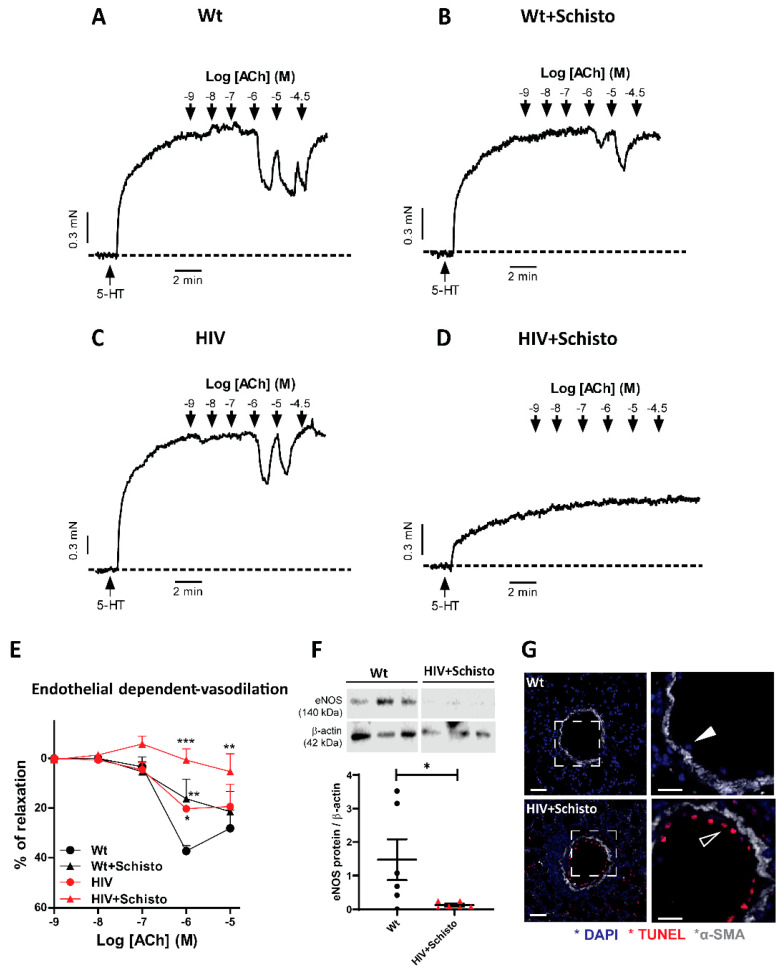
**Endothelial dysfunction in HIV mice is aggravated by Schistosoma egg exposure.** (**A**–**D**) Original recordings of the relaxation induced by Acetylcholine (ACh) in serotonin-stimulated PA from Wt and HIV mice exposed and unexposed to Schistosoma. (**E**) Relaxant effects of the endothelium-dependent vasodilator ACh in serotonin-stimulated PA. (**F**) Lung endothelial NO synthase (eNOS) protein expression. (**G**). Representative immunofluorescence images of TUNEL (red) and α-SMA (gray) staining in lungs of Wt (upper panel) and HIV+Schisto (lower panel) mice, including higher magnifications (white squares) of TUNEL-negative (close arrowhead) and TUNEL-positive (open arrowhead). Scale bar 50 µm. Values are mean ± SEM (n = 5–6). * *p* < 0.05, ** *p* < 0.01, and *** *p* < 0.001, as determined by two-way repeated measures ANOVA analysis followed by the Bonferroni’s post-hoc test (panel (**E**)) or Student’s unpaired *t*-test (panel (**F**)).

**Figure 3 cells-11-02414-f003:**
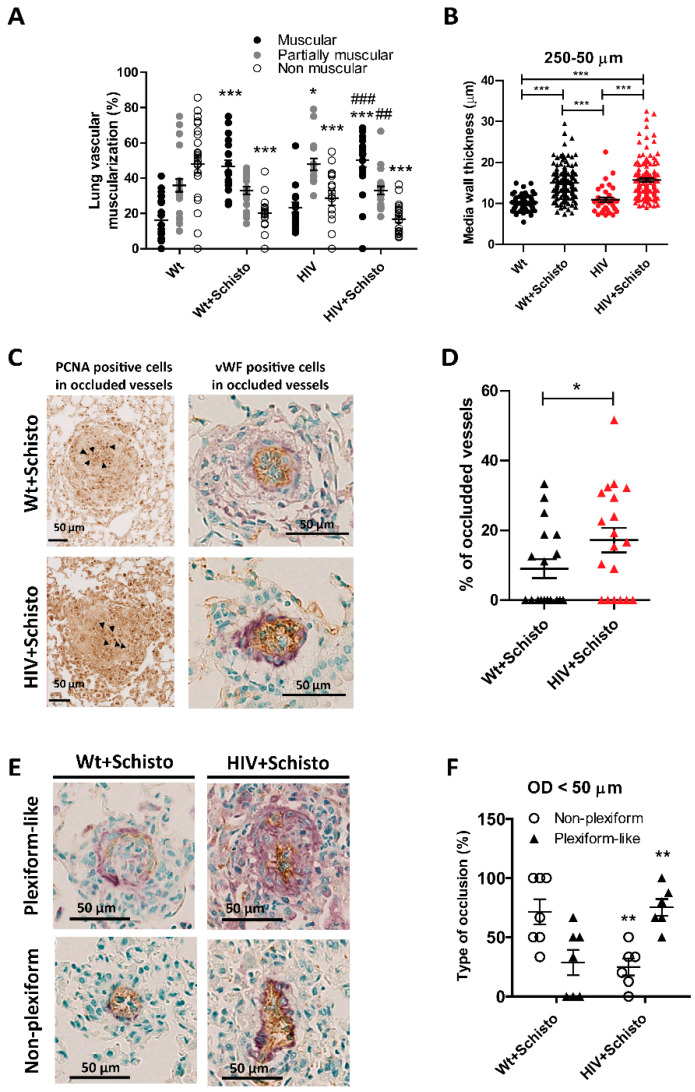
**Co-exposed animals show enhanced obliterative pulmonary vascular remodeling.** (**A**) Percentage of muscular (black), partially muscular (gray), and non-muscular (white) vessels (n = 16–23 mice per group). * and *** indicate *p* < 0.05 and *p* < 0.001, respectively vs. Wt; ^##^ and ^###^ indicate *p* < 0.01 and *p* < 0.001 vs. HIV, respectively. (**B**) Medial wall thickness of PA between 250 and 50 μm. Between 35 to 100 pulmonary vessels were analyzed per group (n = 8–13 mice). *** *p* < 0.001. Results are shown as mean ± SEM. One-way ANOVA analysis followed by Tukey post-hoc test was applied. ((**C**), left). Representative images of PCNA staining in occluded lesions (solid arrowheads point to representative PCNA^+^ cells). ((**C**), right) Representative images of occluded lesions where cellular identity was visualized by antibodies against α-smooth muscle actin (purple/violet color) and von Willebrand factor (vWF, brown color) in Wt (upper panel) and HIV (lower panel) mice exposed to Schistosoma eggs. Results are expressed as mean ± SEM. * *p* < 0.05 and *** *p* < 0.001 (one-way ANOVA analysis followed by Tukey post-hoc test). (**D**) Percentage of occluded vessels in Wt and HIV mice exposed to Schistosoma eggs (n = 19 mice per group). (**E**) Representative cross-sectional views of α-smooth muscle actin (purple/violet color) and von Willebrand factor (vWF, brown color) staining of plexiform-like (upper panels) and non-plexiform-like type (bottom panels) of occluded vessels (magnification 40x; scale bar 50 µm). (**F**) Percentage of severe occluded vessels of outer diameter <50 µm. Results are expressed as mean ± SEM. * *p* < 0.05 and ** *p* < 0.01 versus Wt+Schisto determined by unpaired Student’s *t*-test.

**Figure 4 cells-11-02414-f004:**
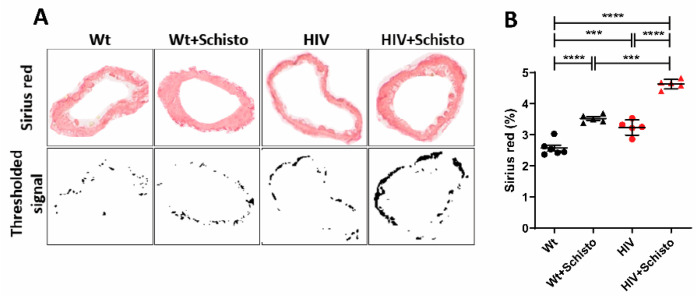
**HIV mice show augmented pulmonary perivascular fibrosis with marked exacerbation after exposure to schistosome eggs**. (**A**) Representative images of pulmonary vessels stained with Sirius red (upper panels). Stained vessel areas are shown as thresholded signal (botton panels). (**B**) Quantification of Sirius red staining as percent Sirius red-positive area fraction (as shown in (**A**), bottom) in the vessel area analyzed. Mice and vessels analyzed per group were: Wt (6, 77), Wt+Schisto (5, 83), HIV (5, 53), and HIV+Schisto (5, 75). Results are expressed as mean ± SEM. *** *p* < 0.001 and **** *p* < 0.0001 as determined by one-way ANOVA analysis followed by Tukey’s post-hoc test.

**Figure 5 cells-11-02414-f005:**
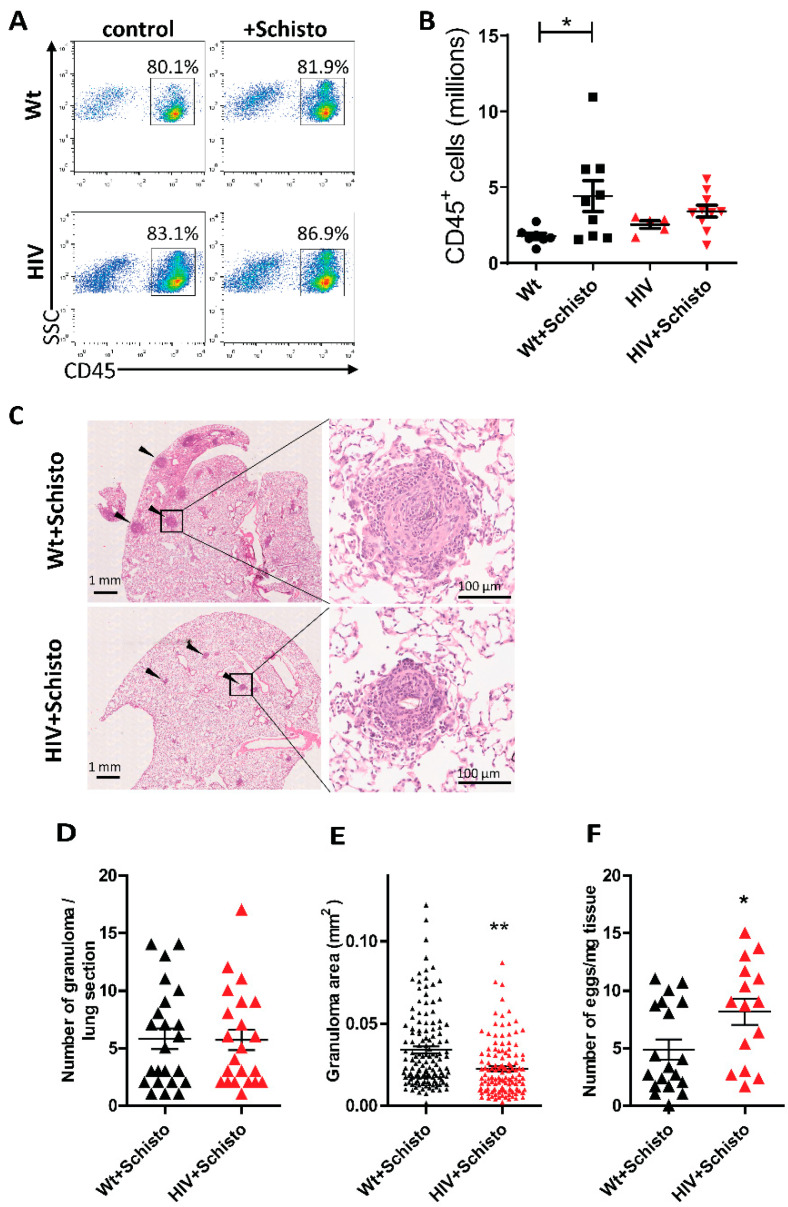
**Reduced pulmonary leukocyte infiltration, granuloma size, and increased egg burden in HIV mice after schistosome egg exposure.** (**A**) Flow cytometric analysis of leukocytes (CD45^+^ cells) in the lung of Wt and HIV mice unexposed (control) or exposed to Schistosoma eggs (+Schisto). SSC indicates side scatter. Numbers within dot plots indicate the percentage of cells in the gated region. (**B**) Numbers of leukocytes (n =5–11). * *p* < 0.05 (one-way ANOVA) (**C**) Left: representative images showing lung granulomas (solid arrowheads) in hematoxylin-and-eosin-stained lung sections from Wt and HIV mice exposed to Schistosoma eggs. Right: magnification of representative images of a parenchymal peri-egg granuloma. (**D**) Number of granulomas per lung section (n = 22–23 mice), (**E**) granuloma area (126 and 134 granulomas analyzed from n = 22–23 mice, respectively), and (**F**) number of eggs recovered from lung digests in Wt (n = 15) and HIV (n = 19) mice exposed to Schistosoma eggs. Results are shown as mean ± SEM. * *p* < 0.05 and ** *p* < 0.01 (Student’s unpaired *t*-test).

**Figure 6 cells-11-02414-f006:**
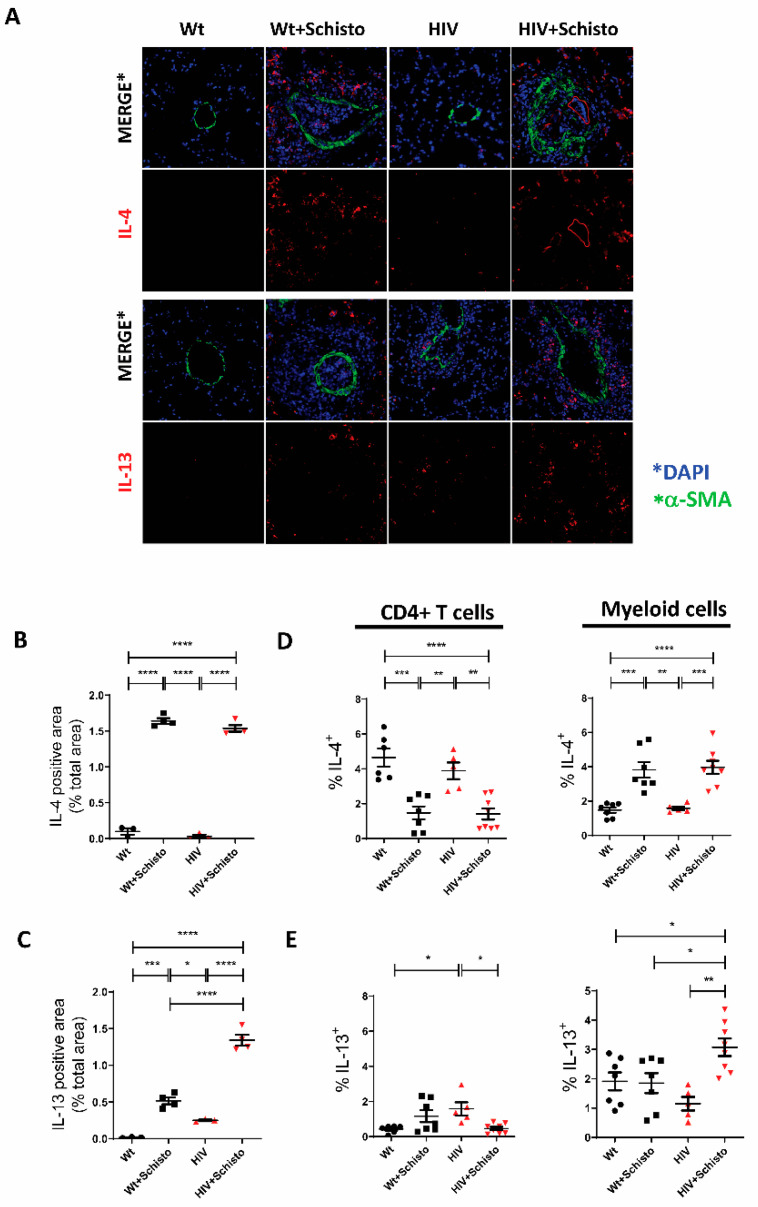
**Altered IL-13 but not IL-4 expression in the lung of mice co-exposed to HIV and Schistosoma** (**A**) Representative immunofluorescence images of lung sections stained with antibodies to α-SMA (green) as vessel marker, IL-4 and IL-13 (red), and DAPI (blue) to detect cell nuclei. (**B**) Quantification of IL-4 and (**C**) IL-13-positive areas in the sections shown in panel A (n = 3–4 mice per group and 21–31 images per group). * *p* ≤ 0.05, ** *p* ≤ 0.01, *** *p* ≤ 0.001, **** *p* ≤ 0.0001 (one-way ANOVA). Flow cytometric analysis of intracellular (**D**) IL-4 and (**E**) IL-13 expression in CD4^+^ T cells (left panels) and myeloid cells (right panels) (n = 5–8 mice). * *p* ≤ 0.05, ** *p* ≤ 0.01, *** *p* ≤ 0.001, and **** *p* ≤ 0.0001 (one-way ANOVA Tukey).

**Figure 7 cells-11-02414-f007:**
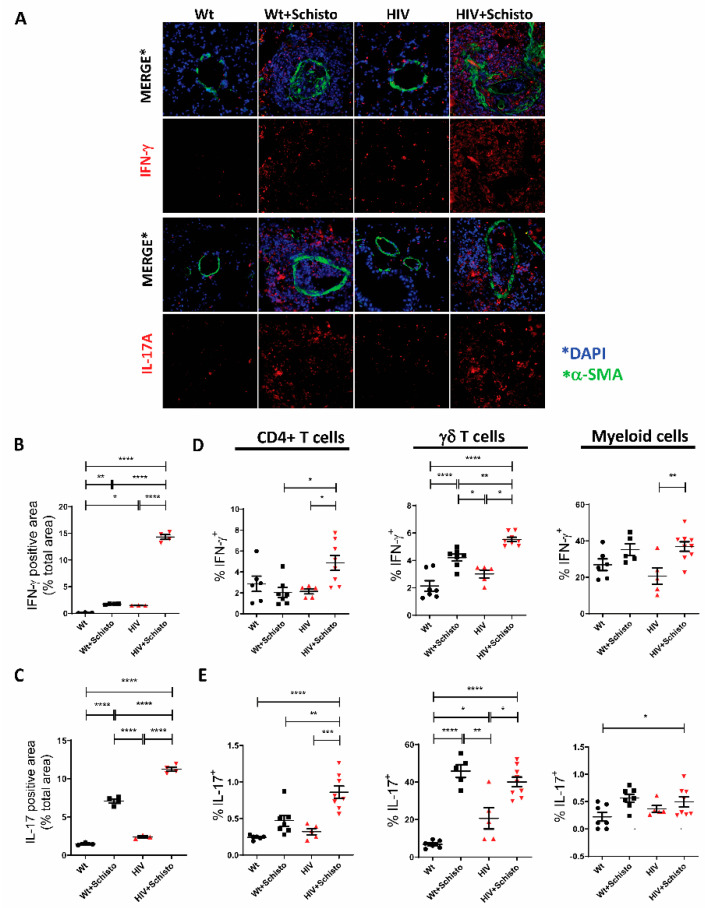
**Pulmonary IFN-γ and IL-17 expression is enhanced in HIV and Schistosoma co-exposed mice** (**A**) Representative immunofluorescence images of lung sections stained with antibodies to α-SMA (green), IFN-γ and IL-17 (**A**) (red), and DAPI (blue). (**B**) Quantification of IFN-γ and (**C**) IL-17A-positive area in the sections shown in panel (**A**) (n = 3–4 mice per group and 21–31 images per group). * *p* ≤ 0.05, ** *p* ≤ 0.01, *** *p* ≤ 0.001, **** *p* ≤ 0.0001 (one-way ANOVA). Flow cytometric analysis of intracellular (**D**) IFN-γ and (**E**) IL-17A expression in CD4^+^ T cells (**left** panel), γδ T cells (**middle** panel), and myeloid cells (**right** panel) (n = 5–8). * *p* ≤ 0.05, ** *p* ≤ 0.01, *** *p* ≤ 0.001, and **** *p* ≤ 0.0001 (one-way ANOVA Tukey).

## Data Availability

Not applicable.

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
