# Peer review of "HIV and Schistosoma Co-Exposure Leads to Exacerbated Pulmonary Endothelial Remodeling and Dysfunction Associated with Altered Cytokine Landscape"

_cells, 2022, doi:10.3390/cells11152414_

Round 1

Reviewer 1 Report

This manuscript details the characterization of an experimental model for the pulmonary endothelial disease under the framework of HIV/Schistosoma co-exposure. While the two pathogens are recognized sources of pulmonary vascular disease, informative reports of the effects of their combination remain lacking in the field.

The proposed model should advance the field tremendously, as it interfaces with PVD mediated by HIV and Schistosoma infections, with recognized global relevance. 

The authors utilized an HIV-transgenic mouse model (Tg26) in which they embolized Schistosoma eggs to the lungs. This approach seems highly reproducible and safe, considering biosafety issues with infectious models. The investigators used male animals aged 9-10 weeks. Overall, Medrano’s results show that the combinations of HIV/Schistosoma elevated both mPAP and RVSP significantly, with reduced Fulton index in this non-infectious model. 

Methods are well detailed, sound, and maximized the use of research animals by measuring mPAP, right ventricular hypertrophy by Fulton, use of myograph to determine vasodilatory capacity, cell immunophenotyping, medial thickness by histology & immunohistochemistry, scoring of lesions, and collagen deposition studies. They also found impaired endothelial-dependent vasodilation in Schistosoma/HIV mice, as well as plexiform-line lesions resulting from abnormal endothelial cell proliferation. The latter is an important observation because, as the authors discuss, this is not a prevalent observation in mice. A high degree of fibrosis was observed in this model. Although the role of HIV infection in the pathology of this model cannot be determined due to the transgenic nature of the model, it reflects the reality of HIV people with viral replication suppressed by antiviral therapy. Statistical analyses seem appropriate for the study.

Based on the apparent high feasibility, it should be easy for other investigators to reproduce this model.

Minor comments: 

-       Please justify why females were not used in this study. 

Author Response

Referee 1

This manuscript details the characterization of an experimental model for the pulmonary endothelial disease under the framework of HIV/Schistosoma co-exposure. While the two pathogens are recognized sources of pulmonary vascular disease, informative reports of the effects of their combination remain lacking in the field.

The proposed model should advance the field tremendously, as it interfaces with PVD mediated by HIV and Schistosoma infections, with recognized global relevance. 

The authors utilized an HIV-transgenic mouse model (Tg26) in which they embolized Schistosoma eggs to the lungs. This approach seems highly reproducible and safe, considering biosafety issues with infectious models. The investigators used male animals aged 9-10 weeks. Overall, Medrano’s results show that the combinations of HIV/Schistosoma elevated both mPAP and RVSP significantly, with reduced Fulton index in this non-infectious model. 

Methods are well detailed, sound, and maximized the use of research animals by measuring mPAP, right ventricular hypertrophy by Fulton, use of myograph to determine vasodilatory capacity, cell immunophenotyping, medial thickness by histology & immunohistochemistry, scoring of lesions, and collagen deposition studies. They also found impaired endothelial-dependent vasodilation in Schistosoma/HIV mice, as well as plexiform-line lesions resulting from abnormal endothelial cell proliferation. The latter is an important observation because, as the authors discuss, this is not a prevalent observation in mice. A high degree of fibrosis was observed in this model. Although the role of HIV infection in the pathology of this model cannot be determined due to the transgenic nature of the model, it reflects the reality of HIV people with viral replication suppressed by antiviral therapy. Statistical analyses seem appropriate for the study.

Based on the apparent high feasibility, it should be easy for other investigators to reproduce this model.

We would like to acknowledge the referee´s constructive and positive comments on our manuscript.

Minor comments: 

-       Please justify why females were not used in this study. 

The point raised by the referee is of great relevance in the context of a disease like pulmonary arterial hypertension (PAH). In fact, PAH is more frequent in females than in males, but females develop a somewhat milder form of PAH. This sex paradox has been attributed to the effects of sex hormones together as well as to additional potential mechanisms. This sex paradox has been commonly attributed to the effects of sex hormones (PMID: 33541615, 24816487), although additional mechanisms have been suggested (PMID: 24816487, 29880496, 29545367).

In our study, the need to compare 4 experimental groups (Wt, Wt+Schisto, HIV and HIV+Schisto) together with the limitations of time and resources made it unfeasible to perform the analysis in both sexes as a starting point. While gender differences in the immune response associated to PAH have been reported (i.e.; PMID: 24816487, 26421302, 29545367), administration of Schistosoma mansoni eggs seems to induce a comparable immune response with an associated granuloma formation and pulmonary vascular remodelling in male (PMID: 31259624) and female (PMID: 28680583, 31339057) mice. Nevertheless, it would be very interesting to evaluate if HIV and Schistosoma co-exposure exerts a differential impact in females compared to the present study in males. The following has been included in the Discussion: “Our study has the limitation of examining the co-exposure to HIV and Schistosoma only in males. Although the administration of Schistosoma mansoni eggs seems to induce a comparable immune response with an associated granuloma formation and pulmonary vascular remodelling in male (PMID: 31259624) and female (PMID: 28680583, 31339057) mice, it would be interesting to evaluate if combined exposure to HIV and Schistosoma exerts a differential impact in females as compared to the present study in males.”

Reviewer 2 Report

This study investigates the effects of HIV and Schistosoma co-exposure on the development of pulmonary endothelial remodeling and function associated with altered cytokine landscape.  It is a well-designed study and a well-written manuscript. The authors used an animal model based on lung embolization of Schistosoma mansoni eggs in HIV-1 transgenic mice. Interestingly, the authors found that co-exposure exacerbates pulmonary endothelial remodeling and impairs endothelial-dependent vasodilation. finally, the authors also found that HIV mice displayed an impaired immune response to parasite eggs in the lungs as evidenced by leukocyte infiltration and increased expression of pro-inflammatory/pro-fibrotic cytokines. This is a very interesting study, and the experiments are technically sound and very well-conducted. This manuscript reports valuable findings and would be of great interest to the field. Nonetheless, a few points should be addressed, at least in part:

-          1. In Fig.1, RVSP  is not significantly affected by the co-exposure while the PAP seems significantly increased in HIV/Schisto compared to WT+schisto. Could the authors comment on these differences? Similarly, media wall thickness is significantly increased in WT+Schisto while PAP isn’t increased. Could the authors also comment on these results?

-          2. In Figure 2, could the authors clarify what is being compared? Did the authors plot the average media thickness per mouse or individual arteries?

-          3. Could the authors measure the activity of eNOS in the lung samples from the mouse model? This would strongly confirm the involvement of eNOS suggested in Figure 2.

-          4. Co-exposed animals show enhanced obliterative pulmonary vascular remodeling. Co-staining PCNA with alpha-SMC or vWF would be useful to confirm whether co-exposure to HIV and Schistosoma affects primarily SMC or EC, or both.

-         5.  Could the authors measure apoptosis levels in SMC and EC in the lung sections?

-          6. Finally, a better characterization of the immunological profile is critical to further strengthen their results. I greatly recommend using the Proteome Profiler Mouse XL Cytokine Array. https://www.bio-techne.com/p/antibody-arrays/proteome-profiler-mouse-xl-cytokine-array_ary028

Author Response

This study investigates the effects of HIV and Schistosoma co-exposure on the development of pulmonary endothelial remodeling and function associated with altered cytokine landscape.  It is a well-designed study and a well-written manuscript. The authors used an animal model based on lung embolization of Schistosoma mansoni eggs in HIV-1 transgenic mice. Interestingly, the authors found that co-exposure exacerbates pulmonary endothelial remodeling and impairs endothelial-dependent vasodilation. finally, the authors also found that HIV mice displayed an impaired immune response to parasite eggs in the lungs as evidenced by leukocyte infiltration and increased expression of pro-inflammatory/pro-fibrotic cytokines. This is a very interesting study, and the experiments are technically sound and very well-conducted. This manuscript reports valuable findings and would be of great interest to the field. Nonetheless, a few points should be addressed, at least in part:

RESPONSE: We would like to thank the referee for the encouraging characterization of our work, and for providing many helpful comments and suggestions.

-          1. In Fig.1, RVSP  is not significantly affected by the co-exposure while the PAP seems significantly increased in HIV/Schisto compared to WT+schisto. Could the authors comment on these differences? Similarly, media wall thickness is significantly increased in WT+Schisto while PAP isn’t increased. Could the authors also comment on these results?

RESPONSE:

Regarding the first question, in our previous version we included data on RVSP and mean PAP. Following the referee´s comment we have now included two new panels in Figure 1 illustrating hemodynamic data of systolic and diastolic pulmonary arterial pressures (new panels B and C, respectively). As it can be observed values of RVSP and systolic PAP are comparable in the different groups. We agree with the referee´s observation that there is a tendency of increased RVSP, SPAP and especially dPAP and mPAP in HIV/Schisto compared to WT+Schisto, but these differences did not reach statistical significance. The following has been added in the results section “Untreated Wt and HIV mice had similar RVSP, systolic, diastolic and mean PAP” and “Egg-treated mice showed a general tendency to higher values of these hemodynamic parameters, with the differences being significant for diastolic and mean PAP in egg-treated HIV mice compared to untreated ones.”

Regarding the second question, we mentioned this issue in our previous version but have further commented on this in the new version as suggested. The paragraph regarding this issue in the discussion reads as follows: “In our model, unlike some previous studies [[26, 28, 29]], lung embolization of schistosome eggs was not associated with overt pulmonary hypertension or cardiac remodeling. Other studies have also found no significant elevation of RVSP or RV hypertrophy in mouse models of experimental schistosomiasis [[14, 15]] even when, as in our study, an increased muscularization of small pulmonary vessels, thickening of the medial layer and even occasional plexiform-like lesions could be observed [[14]]. Differences in experimental conditions such as altitude (Denver vs Madrid) or mouse genetic background (C57BL/6 vs FVB) [70] could explain discrepancies among studies.”

-          2. In Figure 2, could the authors clarify what is being compared? Did the authors plot the average media thickness per mouse or individual arteries?

RESPONSE: We apologise that our data presentation failed to show clearly the analysis performed. In the revised manuscript we have clarified this issue and added the following information in the figure legend. “Between 35 to 100 pulmonary vessels were analyzed per group (n=8-13 mice)”.

-          3. Could the authors measure the activity of eNOS in the lung samples from the mouse model? This would strongly confirm the involvement of eNOS suggested in Figure 2.

RESPONSE: We appreciate this comment. Following your recommendation, we have now measured eNOS expression in lung samples. Western blot analysis shows that animals co-exposed to HIV plus Schistosoma have a marked reduction of eNOS expression that correlates with the impaired endothelial-dependent vasodilation observed. These new data are shown in Figure 2F and referred to in Results “Consistent with our functional data we found that the expression of eNOS, the main source of endothelial NO, was markedly reduced in lungs from co-exposed mice compared to Wt (Figure 2F)”  and Discussion “This was associated with blunted eNOS expression and with increased apoptosis in the inner layer of PA from co-exposed animals”

-          4. Co-exposed animals show enhanced obliterative pulmonary vascular remodeling. Co-staining PCNA with alpha-SMC or vWF would be useful to confirm whether co-exposure to HIV and Schistosoma affects primarily SMC or EC, or both.

RESPONSE: Following the Referee´s suggestion, we performed PCNA, alpha-SMA (SMC) and ERG (as a marker of EC) co-staining in lung samples and included representative images in Supplementary Figure 6C. The figure shows that similar to Wt vessels, non occluded vessels from co-exposed animals are essentially PCNA negative. In contrast, vessels that have started the occlusion process show high staining for PCNA. It should be noted that PCNA-positive staining was observed in ERG+ and other unidentified cells (probably immune cells) but not in alpha-SMA+ cells in partially occluded vessels. Thus our data suggest that co-exposure to HIV and Schistosoma affects primarily EC to trigger vessel occlusion.

The following has been added in the Results “To gain more information on this, we performed a PCNA, α-SMA (as a marker of smooth muscle cells) and ERG (as a marker of endothelial cells) immunofluorescence co-staining in lung samples from co-exposed animals. Representative images in Supplementary Figure 6C show that while non-occluded vessels from Wt or co-exposed animals are essentially PCNA-negative, partially occluded vessels in HIV+Schisto mice showed high PCNA-positive staining which correspond to endothelial and other unidentified cells (probably immune cells) but not to smooth muscle cells.”.

-         5.  Could the authors measure apoptosis levels in SMC and EC in the lung sections?

RESPONSE: As recommended by the Referee, we performed TUNEL for measuring apoptosis in the lung sections, in combination with α-SMA and CD31/vWF staining for identification of SMC and EC, respectively. Unfortunately, the antibodies to CD31 or vWF did not work in the TUNEL setting, even using different antigen retrieval protocols. To any extent, the results showed that TUNEL+ cells were located in the intima but not in the media (α-SMA+)  layer (new panel G in Figure 2) of pulmonary vessels in mice co-exposed to HIV and Schistosoma, in contrast with WT counterparts where TUNEL staining was practically undetectable. These data reinforce the endothelial-dependent vasodilation impairment observed and further confirm that vascular EC are the primary target of the deleterious effects of combined HIV and Schistosoma co-exposure.

The following has been added in the Results “We also assessed if pulmonary vascular dysfunction in co-exposed mice was associated with increased apoptosis by examining TUNEL staining in lung sections. A clear TUNEL positive staining was observed in the inner, but not in the media, layer of PA from egg-treated HIV mice, while this was not found in Wt mice (Figure 2G)”

The following has been added in the Discussion “In line with this notion, we observed increased apoptosis in the tunica intima of non-occluded vessels from egg-treated HIV mice; which suggest that endothelial apoptosis may be the initial trigger for vessel obliteration.”

-          6. Finally, a better characterization of the immunological profile is critical to further strengthen their results. I greatly recommend using the Proteome Profiler Mouse XL Cytokine Array. https://www.bio-techne.com/p/antibody-arrays/proteome-profiler-mouse-xl-cytokine-array_ary028

RESPONSE: While we certainly agree that using a multiplex array to detect multiple cytokines could help to strengthen our study, a limited availability of samples precluded a more comprehensive analysis of pulmonary cytokine expression in the HIV/Schistosoma co-exposure model. Because a potential shortage of samples was somehow foreseen at the inception of the study, we decided to focus on cytokines previously shown to be critically involved in the pulmonary pathology following individual exposure to Schistosoma eggs or HIV, namely IL-4, IL-13, IFN-g and IL-17. The idea was to have a minimal framework for comparison with the novel co-exposure situation, by using complementary approaches (flow cytometry and immunofluorescence microscopy) for improving consistency of the results.

In an independent set of experiments, a comprehensive flow cytometric surface phenotyping of pulmonary immune cells was performed in HIV transgenics and WT counterparts, exposed or not to Schistosoma eggs. However, we feel that the data generated is too large to be incorporated in the present manuscript and relatively out of its scope. We aim to prepare a specific manuscript describing the phenotyping of pulmonary immune cells in our model in the near future.

Round 2

Reviewer 2 Report

The authors have addressed all my questions and concerns. No further comments. Good job!